# Osteoarticular Infections in Children: Accurately Distinguishing between MSSA and *Kingella kingae*

**DOI:** 10.3390/microorganisms11010011

**Published:** 2022-12-20

**Authors:** Benoit Coulin, Giacomo DeMarco, Oscar Vazquez, Vasiliki Spyropoulou, Nathaly Gavira, Tanguy Vendeuvre, Anne Tabard-Fougère, Romain Dayer, Christina Steiger, Dimitri Ceroni

**Affiliations:** Paediatric Orthopaedics Service, Geneva Children’s Hospital, Geneva University Hospitals, 1211 Geneva, Switzerland

**Keywords:** osteoarticular, infection, osteomyelitis, septic arthritis, MSSA, *Kingella kingae*

## Abstract

Introduction: Osteoarticular infections (OAIs) constitute serious paediatric conditions that may cause severe complications. Identifying the causative organism is one of the mainstays of the care process, since its detection will confirm the diagnosis, enable adjustments to antibiotic therapy and thus optimize outcomes. Two bacteria account for the majority of OAIs before 16 years of age: *Staphylococcus aureus* is known for affecting the older child, whereas *Kingella kingae* affects infants and children younger than 4 years old. We aimed to better define clinical characteristic and biological criteria for prompt diagnosis and discrimination between these two OAI. Materials and methods: We retrospectively studied 335 children, gathering 100 *K. kingae* and 116 *S. aureus* bacteriologically proven OAIs. Age, gender, temperature at admission, involved bone or joint, and laboratory data including bacterial cultures were collected for analysis. Comparisons between patients with OAI due to *K. kingae* and those with OAI due to *S. aureus* were performed using the Mann–Whitney and Kruskal–Wallis tests. Six cut-off discrimination criteria (age, admission’s T°, WBC, CRP, ESR and platelet count) were defined, and their respective ability to differentiate between OAI patients due to *K. kingae* versus those with *S. aureus* was assessed by nonparametric receiver operating characteristic (ROC) curves. Results: Univariate analysis demonstrated significant differences between the two populations for age of patients, temperature at admission, CRP, ESR, WBC, and platelet count. AUC assessed by ROC curves demonstrated an exquisite ability to discriminate between the two populations for age of the patients; whereas AUC for CRP (0.79), temperature at admission (0.76), and platelet count (0.76) indicated a fair accuracy to discriminate between the two populations. Accuracy to discriminate between the two subgroups of patients was considered as poor for WBC (AUC = 0.62), and failed for ESR (AUC = 0.58). On the basis of our results, the best model to predict *K. kingae* OAI included of the following cut-offs for each parameter: age < 43 months, temperature at admission < 37.9 °C, CRP < 32.5 mg/L, and platelet count > 361,500/mm^3^. Conclusions: OAI caused by *K. kingae* affects primarily infants and toddlers aged less than 4 years, whereas most of the children with OAI due to MSSA were aged 4 years and more. Considering our experience on the ground, only three variables were very suggestive of an OAI caused by *K. kingae*, i.e., age of less than 4 years, platelet count > 400,000, and a CRP level below 32.5 mg/L, whereas WBC and ESR were relatively of limited use in clinical practice.

## 1. Introduction

Osteoarticular infections (OAIs) constitute serious paediatric conditions that may evolve into major morbidity, including disruptions to subsequent bone development, joint destruction, and permanent articular disability [1]. Identifying the causative organism is one of the mainstays of the care process since its discovery will confirm the diagnosis, enable adjustments to antibiotic therapy and thus optimize outcomes [2,3]. A variety of causative pathogens have been identified, and *S. aureus* was long considered the most common microorganism in every paediatric age category [3,4]. However, recent improvements in our knowledge about the aetiologies and epidemiology of OAIs in paediatric populations have shown this view to be too simplistic. The routine widespread use of molecular methods of pathogen identification has revealed significant modifications from the previously expected microbial epidemiology [5,6]. Understanding of the microbiological causes of OAI has evolved significantly in recent years, and their clinical and biological characteristics are currently considered to be closely correlated to children’s ages and the causative pathogens. Indeed, it is now recognised that the pathogens responsible for paediatric OAIs depend not only on the child’s age but also on their comorbidities, immune and vaccination statuses, socioeconomic conditions, changes in patterns of immunomodulating diseases, and the emergence of resistant bacteria [3]. In the last decade, due to the extensive use of nucleic acid amplification assays in diagnostic processes, *K. kingae* has emerged as the most frequently isolated pathogen in cases of OAI [5,6,7,8,9].

OAIs caused by *K. kingae* are considered to be benign, with a mild-to-moderate clinical presentation, a favourable prognosis after antibiotic treatment and few long-term sequelae [9,10,11,12,13,14]. These features are partly explained by *K. kingae*’s low virulence [10] and partly by its high susceptibility to β-lactam antibiotics [10,15,16]. Contrarily, pyogenic OAIs caused by aggressive pathogens—such as methicillin-sensitive/resistant S. aureus (MSSA and MRSA), streptococci, or Gram-negative organisms—have a more dramatic presentation, and the children affected often appear quite ill, presenting with high fever and an elevated white blood cell (WBC) count [17,18]. These infections must be considered as orthopaedic emergencies with the potential for such adverse outcomes as systemic sepsis, cartilage destruction, growth cartilage damage and avascular necrosis, which could lead to severe impairments in bone development and irreversible loss of function [1]. It, therefore, becomes essential, in our opinion, to establish criteria to distinguish these two clinical entities, since they entail different degrees of urgency for therapeutic management and of need for a surgical procedure. The present study aimed to compare the signs and symptoms of disease caused by *K. kingae* and methicillin-susceptible *Staphylococcus aureus* (MSSA) and then establish predictive values to enable the differentiation of OAIs induced by these two organisms.

## 2. Materials and Methods

After approval from the Children’s Hospital Ethics Review Committee (CE 14-102R), we retrospectively reviewed the medical charts of all children aged from 0–15 years old who were admitted to our institution for a suspected OAI, from January 2007 for confirmed cases of *K. kingae* and from January 1997 for confirmed cases of *S. aureus*, up until December 2019 (January 2007 corresponding to the implementation of a routine molecular detection method for *K. kingae*). The criteria established by Morrey [19,20,21] and Morrissey [22] were used to estimate children’s risk of having a bone or joint infection. Diagnoses of an OAI were confirmed using imaging techniques (plain radiography, 99 mTC bone scan, magnetic resonance imaging) according to established criteria [23]. Children diagnosed with a musculoskeletal infection were further categorized with the following diagnoses: acute haematogenous osteomyelitis, subacute osteomyelitis, septic arthritis, acute osteomyelitis with concomitant septic arthritis, primary spine infection (spondylodiscitis, vertebral osteomyelitis), septic pyomyositis, septic chondritis and septic tenosynovitis.

Age, gender, temperature at admission, the bone or joint involved, and laboratory data including bacterial cultures (blood, synovial fluid and bone exudate), quantitative polymerase chain reaction (qPCR) assays [24], WBC and differential platelet counts, erythrocyte sedimentation rate (ESR) and serum C-reactive protein (CRP) were all collected for analysis. We used the classic cut-off values for the following four variables, which are considered to have predictive value for infection parameters in clinical practice: fever defined as an oral temperature of ≥38 °C; WBC > 12,000 leukocytes/mm^3^; CRP > 10 mg/L; and ESR > 20 mm/h. Study exclusion criteria included chronic osteomyelitis and infections subsequent to a fracture or surgery. Additive exclusion criteria were also used to avoid information bias associated with incomplete data analysis and selection bias associated with the inclusion of patients with presumptive and inconsistent diagnoses. These were: (i) no bacteriological diagnosis obtained, (ii) no laboratory data available, and (iii) the patient was not eventually managed by the administration of antibiotics. The following two separate diagnostic subgroups were established based on bacteriological identification: OAI due to *K. kingae* and OAI due to *S. aureus*.

### 2.1. Microbiological Methods

Blood cultures have been systematically used for trying to isolate the microorganisms responsible for septic arthritis. This study’s blood culture media were BACTEC 9000 for the period before 2009, and following that, an automated blood culture system (BD BACTEC FX) was used. Joint fluid was sent to the laboratory for Gram staining, cell count, and immediate inoculation onto Columbia blood agar (incubated under anaerobic conditions), CDC anaerobe 5% sheep blood agar (incubated under anaerobic conditions), chocolate agar (incubated in a CO_2_-enriched atmosphere), and brain–heart broth. These media were incubated for 10 days. Two PCR assays were also used for bacterial identification when standard cultures were negative. Initial aliquots (100–200 µL) were stored at −80 °C until processing for DNA extraction. A universal, broad-range PCR amplification of the 16S rRNA gene was performed using BAK11w, BAK2, and BAK533r primers (Eurogentec, Seraing, Belgium). This study also used a real-time PCR assay targeting the *K. kingae* gene’s rtx toxin from 2007 [24]. The assay is designed to detect two independent gene targets from the *K. kingae* rtx toxin locus, namely *rtxA* and *rtxB* [24]. This PCR assay detecting *K. kingae* was used to analyse different biological samples, such as synovial fluid, bone or discal biopsy specimens, or peripheral blood. Since September 2009, we have also been carrying out oropharyngeal swab PCR for children from 6 months to 4 years old. It has been demonstrated that this simple technique for detection of *K. kingae* rtx toxin genes in the oropharynx provides strong evidence that this microorganism is responsible for OAI, or even stronger evidence that it is not [25].

### 2.2. Statistical Analysis

The characteristics of patients with an OAI were analysed in the two diagnostic subgroups. Clinical manifestations and laboratory test results were expressed as median and range, as well as mean and standard deviation. Comparisons between patients with a *K. kingae* OAI and those with an *S. aureus* OAI were performed using Student’s *t*-tests.

We defined six predictors (i.e., age, temperature, WBC, CRP, ESR and platelet count) and used non-parametric receiver operating characteristic (ROC) curves to assess their abilities to differentiate between patients with *K. kingae* OAIs and those with *S. aureus* OAIs. The areas under the ROC curve (AUC) and their 95% confidence intervals were assessed using the nonparametric method. An AUC value of 0.5 indicates that a parameter is worthless, as it is only as effective as a random guess for distinguishing patients with a *K. kingae* OAI from those with an *S. aureus* OAI. An AUC of 1, however, represents a perfect test. A rough guide for classifying the accuracy of a diagnostic test is the traditional academic point system: i.e., 0.90–1 = excellent; 0.80–0.90 = good; 0.70–0.80 = fair; 0.60–0.70 = poor; and 0.50–0.60 = fail. Cut-off values were determined for all six parameters, and only those with a discriminative ability > 0.75 were selected for further analysis. The parameters were then dichotomised and determined for each patient. The proportions of OAIs due to *K. kingae* and *S. aureus* were assessed according to the number of parameters present at admission.

In order to predict, regardless of age, OAIs caused by *K. kingae* in children less than 4 years old, the clinical and laboratory parameters (temperature at admission, CRP, WBC, ESR, platelet count) were then included in a univariate and multivariate logistic regression model, for which adjusted OR and 95% CI were calculated.

Statistical analysis was performed using R v.4.2.2 software (R foundation for statistical computing, Vienne, Austria) with the RStudio interface (RStudio Team 2016, RStudio, Inc., Boston, MA, USA). Statistical significance was set at *p* < 0.05.

## 3. Results

Of 335 eligible patients, 116 had an OAI identified as due to *S. aureus*, and 100 were confirmed as having an OAI caused by *K. kingae*.

### 3.1. OAI Due to K. kingae

In the 12 year period during which *K. kingae* OAIs were examined, 100 patients were considered to be confirmed cases (51 girls, 49 boys). Its presence was confirmed using a positive radiological study (MRI) and at least one proof of a local bacteriological presence. Identification of *K. kingae* was possible through blood PCR (10 cases), in a bone/fluid culture (1 case) and in a bone/joint fluid PCR assay specific for *K. kingae* (97 cases).

The mean age (±SD) of children with a confirmed *K. kingae* OAI was 19.9 ± 12.7 months, ranging from 7–75 months old. The highest prevalence was between 7 and 24 months old (*p* < 0.001), with 80% of patients in this range. (Figure 1). No confirmed *K. kingae* OAIs occurred under 7 months old, whereas there were 4 confirmed cases in children older than 4 years old. With regards to types of OAIs, types and locations of infections are listed in Figure 2. *K. kingae* mostly caused septic arthritis and primarily affected the knee.

The clinical and laboratory parameter distributions are summarized in Table 1. We observed that 76% of patients with a confirmed *K. kingae* OAI were afebrile (T < 38 °C) on admission, but most had presented a fever peak above 38 °C before admission. WBC count was considered elevated in 10% (>17,000/mm^3^ in children younger than 48 months old and >12,000/mm^3^ in older children), whereas no left shift was noted. CRP was considered abnormal (>10 mg/L) in 66% of patients. The mean of the abnormal values was 24 ± 23 mg/L. ESR was abnormal (>20 mm/h) in 70.7% of cases. Finally the platelet count was abnormal (>400,000/mm^3^) in 50% of cases.

### 3.2. OAIs Due to S. aureus

Over 22 years, OAIs due to *S. aureus* were confirmed in 116 patients (38 girls, 78 boys) using positive MRI and positive blood and/or bone/fluid cultures. All of the *S. aureus* infections encountered in this study were MSSA; nevertheless, in 4 cases, OAI was due to Panton-Valentine leukocidin (PVL)-producing MSSA. MSSA were recovered from blood cultures in 56 cases of 98 examinations performed (57.1%). In another 97 cases, the pathogen was identified through bone/fluid cultures (90.7% of all cultures realized). In 16 additional cases, MSSA was also identified by performing PCR assays.

The mean age of children with a confirmed MSSA OAI was 109 ± 51 months old, ranging from 1 to 188 months. The highest prevalence was between 6 and 15 years old (*p* < 0.001), with 72% (84/116) of confirmed MSSA OAIs within this age range. Inside this interval, most affected children were aged from 9 to 12 years (37%). The types and locations of MSSA infections are listed in Figure 2. MSSA mostly caused acute haematogenous osteomyelitis. The most affected bones were in the lower limbs.

The clinical and laboratory parameter distributions are summarized in Table 1. We observed that 63% of patients with a confirmed MSSA OAI were febrile (T ≥ 38 °C) on admission. WBC count was considered elevated in 34 patients, with a band shift in 25 cases. A CRP level was available in 111 cases and was considered elevated (>10 mg/L) in 100 patients (90.1% of patients) and normal in the remaining 11 cases. When considering only the abnormal values, the mean value of CRP was 87 ± 67 mg/L. The ESR was measured in 87 patients and was abnormal (>20 mm/h) in 67 cases (77%). The platelet count was abnormal (>400,000/mm^3^) in only 17% of cases.

### 3.3. Comparison of Demographic Characteristics and Clinical and Biological Features between OAIs Caused by MSSA and K. kingae

Group comparison between *K. kingae* OAIs (n = 100) and *S. aureus* OAIs (n = 116) demonstrated significant differences (*p* < 0.05) in age, temperature at admission, CRP, ESR, and WBC and platelet counts (Table 1).

The AUCs assessed using ROC curves for temperature at admission, CRP and differential WBC count were all significantly different from the non-informative value of 0.5 (Figure 3). The AUC for age at diagnosis demonstrated an excellent ability (AUC = 0.91) to discriminate between the two populations. The AUCs for CRP (0.79), temperature at admission (0.76) and platelet count (0.76) indicated a fair accuracy in discriminating between the two populations. The discriminatory accuracy between the two subgroups of patients was poor for WBC count (AUC = 0.62) and fail for ESR (AUC = 0.58). Therefore, the cut-offs defined at WBC ≥ 9350/mm^3^ and ESR ≥ 51.5 mm/h could not be used as discrimination tools.

As mentioned above, only parameters with an AUC superior to 0.75 provided cut-off values enabling differentiation between *K. kingae* OAIs and *S. aureus* OAIs. Age at admission ≥ 43 months, CRP ≥ 32.5 mg/L, temperature at admission ≥ 37.9 °C and platelet count < 361,500/mm^3^ were highly suggestive for an OAI due to *S. aureus*.

The presence or absence of these parameters was assessed for the two subgroups, and positive and negative predictive values were calculated together with the number of patients above or below these cut-off values (see Table 2). A discriminatory cut-off appeared clearly for age (43 months), and this specific repartition for the two OAIs is clearly illustrated in Figure 1.

Using the four above described discrimination cut-offs (age at admission < 43 months; CRP < 32.5 mg/L; temperature at admission < 37.9 °C; platelet count > 361,500/mm^3^), we clearly observed a discriminative distribution: for confirmed *K. kingae* OAIs, about 38% of children had all four, and 42% had three parameters within the predicted values for a *K. kingae* OAI; 16% had two parameters, 4% had only one, and none had no parameters corresponding to a *K. kingae* infection. On the other side, of the children with confirmed MSSA OAIs, 36% had four parameters within the range predicted for an MSSA infection, 36% had three, and 17.5% had two. Only 10.5% had either one or no clinical or biological parameters in their discrimination levels.

With regard to the four “classical” combined infection predictors (temperature ≥ 38 °C; WBC ≥ 17,000/mm^3^ for children less than 4 years old and ≥12,000/mm^3^ for ≥4 years old; ESR ≥ 20 mm/h; and CRP ≥ 10 mg/L), 15.4% of children with a *K. kingae* OAI had no positive predictor, 25.6% had a single predictor, 41% had two, 14.2% had three, and only 3.8% had all four abnormal values. In comparison, 4.8% of children with an MSSA OAI had no positive predictors, 10.8% had one, 24.1% had two, 42.2% had three, and 18.1% had all four predictors. Interestingly, WBC was rarely elevated in an isolated manner; in *K. kingae* OAIs, an elevated WBC count was always correlated a with an elevated ESR and CRP level. Similarly, an abnormal WBC count in children with an MSSA OAI was associated with 92% of cases with an abnormal CRP level (Figure 4).

### 3.4. Prediction of OAIs Caused by K. kingae with Clinical and Biological Features in Children Less Than 4 Years Old

In children less than 4 years old, univariate logistic regression models demonstrated that CPR (*p* < 0.001), temperature at admission (*p* = 0.001), ESR and WBC count (*p* < 0.05) were significant determinants of *K. kingae* OAI (Table 3). In the multivariate analysis, CRP (OR = 0.996; 95% CI, 0.994–0.998; *p* < 0.001) was the only significant determinant of *K. kingae* OAIs.

## 4. Discussion

Since the 2000s, the increasing use of nucleic acid amplification assays in the diagnostic process for OAIs has significantly increased the ability to detect fastidious pathogens and has changed contemporary bacteriological epidemiology [5,6]. The present study confirms primarily that *K. kingae* and MSSA are currently the two main pathogens responsible for OAIs, and discriminating between them rapidly is essential. In view of their very different clinical course, our results emphasised that age of patients was the main difference between children with a *K.* kingae OAI and those with an *S. aureus* OAI [26]. In fact, children with OAIs caused by *K. kingae* were significantly younger than those affected by infections due to MSSA (mean age 19.9 months vs 9.1 years). Indeed, 96% of children with an OAI caused by *K. kingae* were less than 4 years old, whereas 85.3% of children with an OAI due to MSSA were 4 years old or more, with 50% of them being aged 10 years or older. This confirmed what is now generally accepted, i.e., that invasive *K. kingae* OAIs occurs predominantly in children under 4 years old, and above this age, invasive *K. kingae* infections are exceptional.

Our results also underlined that the clinical presentation of OAIs due to *K. kingae* was very different from those caused by MSSA. In general, young children with *K. kingae* OAIs were admitted with less acute symptoms than the classic presentation of a sick child with fever. We observed that only 24% of children with a *K. kingae* OAI had a temperature > 38 °C at admission, whereas 63% of children with an MSSA OAI had a temperature > 38 °C. Mean temperature at admission was significantly lower in children with a *K. kingae* OAI (37.3 °C) than in children with an *S. aureus* OAI (38.2 °C). In a large, multicentre study of 169 paediatric patients with a culture-proven OAI, Dubnov-Raz et al. reported that children with a *K. kingae* OAI frequently presented with a body temperature < 38 °C [11]. Similarly, Chometon et al. reported that only 13 of 39 patients (33.3%) with a *K. kingae* OAI had a fever (≥38 °C) at admission [27]. In two previous studies, we noted that the mean temperatures at the admission of paediatric patients with a *K. kingae* OAI were 37 °C (30 patients) [26] and 37.2 °C (66 patients) [5], which tends to corroborate the findings of those other authors.

The third point to emerge from our study was that *K. kingae* OAIs are characterised by a more moderate inflammatory biological response than MSSA OAIs. CRP levels were abnormal in 66% of children (66/100) with a *K. kingae* OAI, with a mean level of 24 mg/L. Contrarily, 90.1% (100/111) of children affected by an MSSA OAI had abnormal CRP levels, with the mean CRP level reaching 81.6 mg/L. These values match those published in earlier studies; indeed, CRP levels reached 37 mg/L for septic arthritis and 18 mg/L for osteomyelitis in the largest study ever of paediatric OAIs caused by *K. kingae*, published by Dubnov-Raz et al. [11]. In the same vein, Ilharreborde et al. described a mean CRP level of 39 mg/L in a cohort of 31 children with septic arthritis due to *K. kingae* [12]. As regards OAIs due to MSSA, a large Finnish study (265 children) focusing on OAIs caused by pyogenic pathogens (75.1% were *S. aureus*) demonstrated a mean CRP level of 87 mg/L [28], a level very close to the 81.6 mg/L noted in our study.

We noted that ESRs were abnormal in 75.6% of our children (62/82) with a *K. kingae* OAI, with a mean rate of 32.9 mm/h. Children affected by OAIs due to MSSA maintained a normal ESR in 21% of cases (19/87), and mean ESR reached 44.2 mm/h. Similarly, these rates perfectly matched with those published in earlier studies; in fact, ESRs reached 44.1 mg/L for septic arthritis and 40 mg/L for osteomyelitis in the study published by Dubnov-Raz et al. about infections caused by *K. kingae* [11]. Here too, the mean ESR was also distinctly greater among children with a pyogenic OAI (mainly due to *S. aureus*) than ESRs noted in OAIs due to *K. kingae*.

Finally, WBC counts were only elevated in 10.2% of children with a *K. kingae* OAI, whereas they were elevated in 35.1% of children with an MSSA OAI. Surprisingly, overall mean WBC count was higher for the children with a *K. kingae* OAI (12,300/mm^3^) than for those suffering from an MSSA OAI (11,000/mm^3^). This paradox was due to the fact that WBC counts inferior to 17,000 cells/mm^3^ are considered normal in children below 4 years old. The mean WBC count in children with a *K. kingae* OAI was slightly higher in the Dubnov-Raz study (14,797/mm^3^) [11] than observed in our patients (12,300/mm^3^), in the Chometon study (12,500/mm^3^) [27], or in the Ilharroborde study (12,400/mm^3^) [12], but all of these groups had a mean WBC count inferior to 17,000 cells/mm^3^, which is within the normal range. The large Finnish study (265 children) focusing on OAIs due to pyogenic pathogens (>75% of *S. aureus*) demonstrated a mean WBC count of 12,600/mm^3^ [28]. Thus, there is probably no significant difference in WBC counts between OAIs caused by *K. kingae* or MSSA, when the cut-offs of normality are established according to the patients’ age. It is now unanimously recognised that WBC count is a nonspecific index for inflammation, that it can be normal in as many as 80% of cases, and that its low sensitivity makes it an unreliable indicator of OAIs [29]. However, we should keep in mind that for *K. kingae* infections, an elevated leucocyte count was always correlated with an elevated ESR and CRP level. As for MSSA, in 92% of cases, an abnormally high leucocyte level was associated with an abnormally high CRP level.

Unexpectedly, we noted that the mean platelet count was statistically significantly greater (*p* < 0.001) in children with a *K. kingae* OAI (416,100/mm^3^) than in those with an MSSA OAI (310,900/mm^3^). Thrombocytosis is generally reactive and secondary to an underlying inflammatory condition, such as tissue damage, malignancy or infectious diseases [30]. The definition of thrombocytosis varies between a platelet count >400,000/mm^3^ and >500,000/mm^3^. Thrombopoiesis is usually inhibited after an acute bacterial infection, whereas, in contrast, chronic inflammation is often associated with reactive thrombocytosis [30]. Today, an elevated platelet count is not generally recognised as a diagnostic marker of OAI. Clinical experience has taught us that any infection can present with an elevated platelet count, even in the absence of the classic signs of infection, such as fever or an elevated WBC count. Surprisingly, in our study, reactive thrombocytosis was more often associated with *K. kingae* OAIs than with MSSA infections; this might be explained by the fact that OAIs caused by *K. kingae* generally follow a mild clinical course, probably behaving like a long-standing infection, which can lead to a more sustained reactive thrombocytosis. In our study, we also noted that an elevated platelet count was rarely found without an elevated CRP level or ESR. Thus, our results suggest that an elevated platelet count in children below 4 years old can be considered a good predictive marker of an OAI caused by *K. kingae.*

The present study demonstrated that using the old model of classic predictive values for infection (T > 38 °C; WBC > 12,000/mm^3^; ESR > 20 mm/h; CRP > 10 mg/L) can be useful for discriminating between OAIs caused by *K. kingae* or MSSA. In fact, we clearly observed that MSSA OAIs presented with a more serious clinical picture at admission and should be, therefore, more suspected.

If our previously described cut-offs were added to discriminate between the two bacteria, as expected, the clinical and biological situations mirrored each other. About 35% of all MSSA OAIs had four parameters in the MSSA range, and another 35% had three parameters in that range, whereas of all *K. kingae* infections, 38% and 42% had none or only one parameter in the MSSA range of values, and none was found to have all four parameters outside their cut-off values.

Based on our results, we then attempted to build an algorithm to distinguish between OAIs caused by *K. kingae* and MSSA using the four parameters that demonstrated AUC values above 0.75: i.e., the child’s age, temperature at admission, CRP level and platelet count. The best model for predicting a *K. kingae* OAI includes the following cut-offs for each parameter: age < 43 months, temperature at admission < 37.9 °C, CRP < 32.5 mg/L and platelet count > 361,500/mm^3^. However, only the age parameter demonstrated an excellent ability to discriminate between the two populations; the others (CRP, platelet count and temperature) only showed fair to good accuracy (AUC > 0.75). These results differed somewhat from those we published in 2011 [25]. That earlier study built a combined score to find the best model with which to differentiate *K. kingae* OAIs from those caused by pyogenic pathogens in children below 4 years old. It used the four variables of body temperature < 38 °C, serum CRP < 55 mg/L, WBC < 14,000/mm^3^ and band forms 150/mm^3^ [25]. Although the cut-offs for temperature were fairly similar, there were crucial differences between the two studies for CRP levels and WBC count. Most of the differences between the studies were probably attributable to the fact that the children with OAIs studied by Ceroni et al. were all of a similar age. In the present study, fewer than 15% of children with an MSSA OAI were younger than 4 years old, and 50% of affected children were 10 years old or above. Thus, it is possible that clinical signs and laboratory tests for the diagnosis of bacterial infections may be more significantly different between young and older children, which highlights the importance of assessing paediatric OAIs with particular regard to children’s ages. In a recently published study, Gouveia at al. demonstrated that age ≥ 6 months but ≤2 years, apyrexy and CRP ≤ 100 mg/L was a better model to distinguish *K. kingae* SA from classic pathogens in children <5 years of age, with an overall PPV of 86.7%, 88.6% for *K. kingae* and 83.9% for pyogenic infections [31]. However, it is important to emphasize that the collective of the patients was much smaller in that study (75 children), and that only 11 OAIs were due to MSSA [31].

## 5. Conclusions

The clinical presentation of a *K. kingae* OAI is characterised by mild-to-moderate clinical and biological inflammatory responses to infection, with the consequence that children present few, if any, criteria evocative of an OAI. OAIs caused by *K. kingae* primarily affected infants and toddlers less than 4 years old, whereas 85.3% of children with an OAI due to MSSA were 4 years old or more. Based on our results, the best predictive model for a *K.* kingae OAI includes the following parameter cut-offs: age < 43 months, temperature at admission < 37.9° C, CRP < 32.5 mg/L and platelet count > 361,500/mm^3^. Our clinical experience indicates that only three variables are very suggestive of an OAI caused by *K. kingae*, i.e., age below 4 years old, CRP < 32 mg/L and platelet count >400,000/mm^3^. WBC count and ESR were of relatively little use in clinical practice.

## 6. Clinical Relevance

Our institution’s emergency physicians admitting children with a suspected OAI now possess a model that can rapidly discriminate between infections caused by *K. kingae* and those caused by *S. aureus.* Prompt differentiation between these OAIs is essential because they require significantly distinct therapeutic management. Indeed, OAIs due to *S. aureus* are more aggressive than those caused by *K. kingae*, and they thus require prompter medical and/or surgical treatment.

## Figures and Tables

**Figure 1 microorganisms-11-00011-f001:**
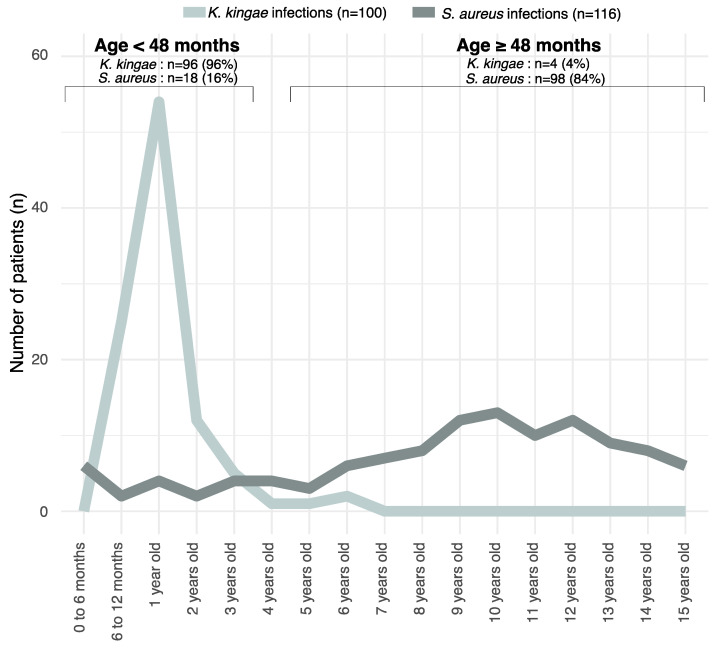
Age at infection for *K. kingae* and *S. aureus*. This chart shows that the incidence of *K. kingae* OAI is greater before 4 years old and almost disappears in older children.

**Figure 2 microorganisms-11-00011-f002:**
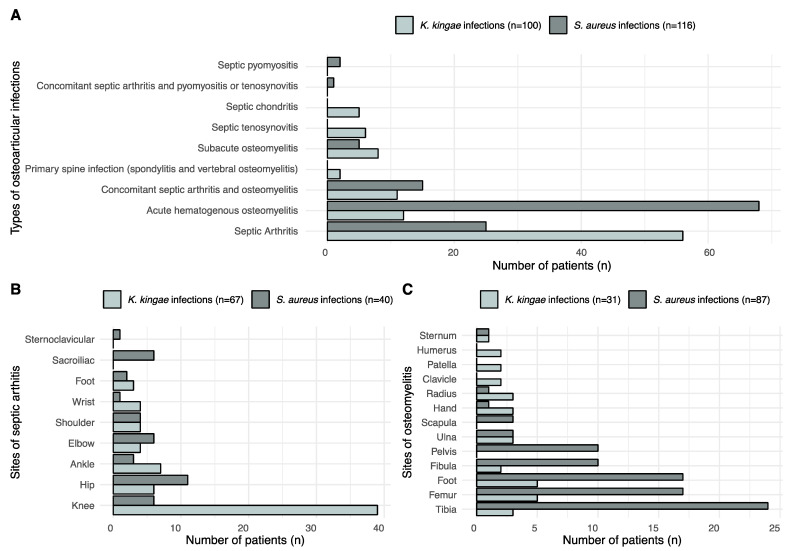
(**A**) Types of infection caused by *S. aureus* and *K. kingae*. We noticed that *S. aureus* mostly caused acute hematogenous osteomyelitis, whereas *K. kingae* caused arthritis. (**B**) Comparison between arthritis sites of *K. kingae* and *S. aureus* infections. We observed that most *K. kingae* arthritis sites involved the knee. *S. aureus* had a more balanced distribution. (**C**) Comparison between osteomyelitis sites of *K. kingae* and *S. aureus* infections. *S. aureus* osteomyelitis mostly affected the tibia and the lower limbs in general.

**Figure 3 microorganisms-11-00011-f003:**
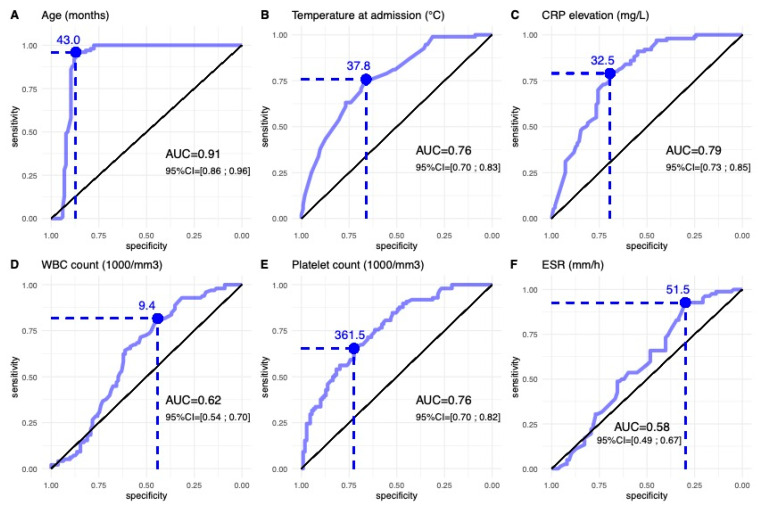
ROC curves for age, temperature at admission, CRP elevation, WBC count, platelet count and ESR to discriminate *K. kingae* OAIs (n = 100) from *S. aureus* OAIs (n = 116). AUC is area under curve, with associated 95% confidence interval (CI).

**Figure 4 microorganisms-11-00011-f004:**
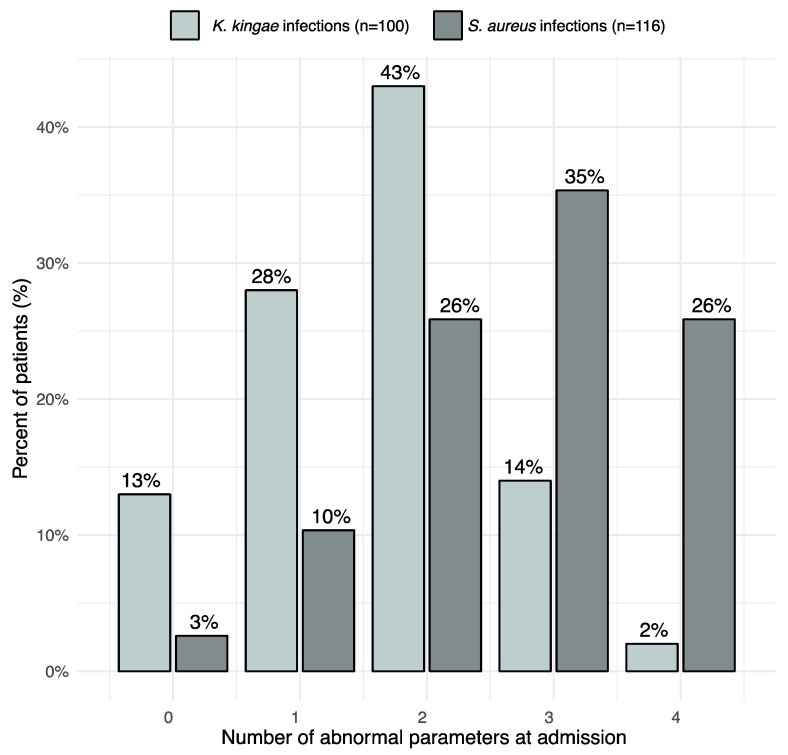
Number of abnormal values present at admission in per cent for the four usual infection predictors (temperature ≥ 38 °C; WBC ≥ 17,000/mm^3^ for children less than 4 years old and ≥12,000/mm for those ≥4 years old; ESR ≥ 20 mm/h; and CRP ≥ 10 mg/L). We observed that most *K. Kingae* OAIs had two or fewer parameters above the normal levels at admission.

**Table 1 microorganisms-11-00011-t001:** Differences in clinical and laboratory variables *between S. aureus* and *K. kingae* germ groups.

	All Patients (n = 216)	*K. kingae* (n = 100)	*S. aureus* (n = 116)	Student *t*-Test		
				*p*	ES	95% CI
Age, mean (SD)	68.0 (58.9)	19.9 (12.7)	109.5 (51.0)	<0.001 *	2.347	−99.4 to −79.9
Temperature, mean (SD)	37.8 (1.1)	37.3 (0.8)	38.2 (1.0)	<0.001 *	1.022	−1.2 to −0.7
CRP elevation, mean (SD)	54.3 (62.7)	24.0 (23.0)	81.6 (73.7)	<0.001 *	1.029	−72.2 to −43.0
WBC count, mean (SD)	11.582 (4.189)	12.300 (3.500)	11.000 (4.600)	0.021 *	0.318	0.2 to 2.4
Platelet count, mean (SD)	360.7 (133.2)	416.100 (115.500)	310.900 (128.700)	<0.001 *	0.860	71.7 to 138.6
ESR, mean (SD)	38.8 (29.0)	32.9 (17.5)	44.2 (35.9)	0.010 *	0.391	−19.9 to −2.7

Statistical tests used were the Student’s *t*-tests when results are presented as mean (SD). Age is in months, temperature is in degrees °C, CRP elevation is in mg/L, WBC and platelet count are /mm^3^, ESR is in mm/h. ES is effect size, CI is confidence interval, *p* is *p*-value. * = *p*-value < 0.05.

**Table 2 microorganisms-11-00011-t002:** Diagnostic ability of optimum cut-offs to discriminate between patients with *S. aureus* and *K. kingae* infections.

		GERMS		Sens./Spec./PPV/NVP (95% CI)
		KK	MSSA	
Age	<43 months	96	15	Sens = 96% (90%; 99%); PPV = 86% (79%; 92%)
	≥43 months	4	101	Spec = 87% (80%; 93%); NPV = 96% (91%; 99%)
Temperature	<37.85 °C	72	38	Sens = 72% (62%; 81%); PPV = 65% (56%; 74%)
	≥37.85 °C	28	78	Spec = 67% (58%; 76%); NPV = 74% (64%; 82%)
CRP elevation	<32.5 mg/L	79	34	Sens = 79% (70%; 87%); PPV = 70% (61%; 78%)
	≥32.5 mg/L	21	82	Spec = 71% (62%; 79%); NPV = 80% (71%; 87%)
WBC count	<9350/mm^3^	82	67	Sens = 82% (73%; 89%); PPV = 55% (47%; 63%)
	≥9350/mm^3^	18	49	Spec = 42% (33%; 52%); NPV = 73% (61%; 83%)
Platelet count	<361,500/mm^3^	66	37	Sens = 66% (56%; 75%); PPV = 64% (54%; 73%)
	≥361,500/mm^3^	34	79	Spec = 68% (59%; 76%); NPV = 70% (61%; 78%)
ESR	<51.5 mm/h	76	61	Sens = 76% (66%; 84%); PPV = 55% (47%; 64%)
	≥51.5 mm/h	34	41	Spec = 47% (38%; 57%); NPV = 70% (58%; 79%)

SENS is sensitivity, SPEC is specificity, PPV is positive predictive value, NPV is negative predictive value.

**Table 3 microorganisms-11-00011-t003:** Univariate and multivariable logistic regression with the biological and clinical markers in patients aged < 48 months to discriminate *S. aureus* OAIs (n = 18) from *K. kingae* OAIs (n = 96).

Predictors	Contrast	Univariate Logistic Regression		Multivariate Logistic Regression	
		OR (95% CI)	*p* Value	OR (95% CI)	*p* Value
CRP	1 mg/L increase	0.995 (0.993–0.996)	<0.001	0.996 (0.994–0.998)	<0.001
ESR	1 mm/h increase	0.997 (0.993–1.000)	0.048	0.998 (0.995–1.000)	0.189
WBC count	1000/mm^3^ increase	0.981 (0.963–0.999)	0.041	0.982 (0.964–1.000)	0.075
Platelet count	1000/mm increase	1.000(0.999–1.000)	0.590	1.000 (1.000–1.000)	0.145
T. admission	1 °C increase	0.876 (0.812–0.946)	0.001	0.962 (0.885–1.040)	0.360
Intercept	-	-	-	12.8 (0.6–279.0)	0.109

OR = Odds Ratio; CI = Confidence Interval.

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
