# Peer review of "Osteoarticular Infections in Children: Accurately Distinguishing between MSSA and Kingella kingae"

_microorganisms, 2022, doi:10.3390/microorganisms11010011_

Round 1
Reviewer 1 Report
The article by Ceroni et al. compares the clinical and laboratory presentation of pediatric skeletal system infections caused by two of the most common etiologies of the disease: methicillin-susceptible Staphylococcus aureus and Kingella kingae. The results clearly show that, with a small overlapping, the two pathogens exhibit consistent differences in age at presentation and inflammatory markers that may assist in establishing a correct presumptive diagnosis, pending culture and/or nucleic acid amplification results. The article confirms in large-scale previous observations [Pediatr. Infect. Dis. J. 2011; 30:902-904.], [Pediatr. Infect. Dis. J. 2011; 30:906-909.], [Pediatr. Infect. Dis. J. 2011; 30:1121-1122.], which suggested that the two etiologies can be distinguished based on the patient's age and acute phase reactants levels but shows that neither ESR nor peripheral blood WBC counts are reliable criteria. The observation that thrombocytosis is especially common in K. kingae osteoarthritis provides added value to the research.
Although the study did not include patients with methicillin-resistant S. aureus because of the severe presentation and clinical course of these infections, it is plausible that the present study results are also relevant to the differentiation between K. kingae and MRSA osteoarthritis.
Major comments
Introduction section, first sentence. In addition to the permanent sequelae of osteomyelitis (“bone development and function”), the authors should mention the long-term disabilities resulting from septic arthritis, such as limping, decreased motion range or ankylosis, joint instability, etc.
Materials and Methods section, page 2, last paragraph. Details of the nucleic acid amplification test should be provided (universal 16S rRNA gene target, K. kingae-specific (rtx operon, groEL, mdh genes?
Materials and Methods section. Small bones and joints and intervertebral discs, which are not easily accessible and rarely subjected to a surgical aspiration or biopsy, were involved in a few patients. Were oropharyngeal specimens obtained for the molecular diagnosis of K. kingae, as advocated by the authors in previous publications [Pediatrics 2013; 131: e230-e235], [J. Child. Orthop. 2010; 4:173-175.], [Pediatr. Infect. Dis. J. 2013; 32:1296-1298]?
Discussion section, page 12, first paragraph, lines 13-15. The authors suggest that the thrombocytosis observed in K. kingae osteoarthritis patients may result from a long-standing infection evolution induced by a low-grade virulence pathogen. Although K. kingae osteomyelitis frequently has an insidious onset and the disease is diagnosed with delay (after 9.2 ± 9.4 days after onset of symptoms), septic arthritis caused by the organism usually has an acute onset, and the infection is suspected after an average of only 3.2 ±3.0 days [Pediatr. Infect. Dis. J. 2010; 29:639-643].
Minor comments
Abstract line 3 and Introduction line 3. Replace "discovery: with "detection".
Introduction: line 13. Replace "pathogens responsible" with "causative pathogens" to avoid repetitions.
Introduction section, last sentence. The article compares the clinical presentation and laboratory markers of only two bacterial species (K. kingae and methicillin-susceptible S. aureus. The sentence should read: "The present study aimed to compare the signs and symptoms of disease caused by K. kingae and methicillin-susceptible Staphylococcus aureus (MSSA) and then establish predictive values to enable the differentiation of OAIs induced by these two organisms".
Figure 1. The actual number of patients in the different age-group categories should be added.
There are inconsistencies in the nomenclature of the two bacterial species. Kingella kingae is referred to as "K. Kingae" (no Italics, Figure 2 and 3) and "K. Kingella" (in Figure 2 Legend), and Staphylococcus aureus as "MSSA", "S. Aureus" (Figure 3, Legend, Figure 3, Legend).
Author Response
The article by Ceroni et al. compares the clinical and laboratory presentation of pediatric skeletal system infections caused by two of the most common etiologies of the disease: methicillin-susceptible Staphylococcus aureus and Kingella kingae. The results clearly show that, with a small overlapping, the two pathogens exhibit consistent differences in age at presentation and inflammatory markers that may assist in establishing a correct presumptive diagnosis, pending culture and/or nucleic acid amplification results. The article confirms in large-scale previous observations [Pediatr. Infect. Dis. J. 2011; 30:902-904.], [Pediatr. Infect. Dis. J. 2011; 30:906-909.], [Pediatr. Infect. Dis. J. 2011; 30:1121-1122.], which suggested that the two etiologies can be distinguished based on the patient's age and acute phase reactants levels but shows that neither ESR nor peripheral blood WBC counts are reliable criteria. The observation that thrombocytosis is especially common in K. kingae osteoarthritis provides added value to the research.
Although the study did not include patients with methicillin-resistant S. aureus because of the severe presentation and clinical course of these infections, it is plausible that the present study results are also relevant to the differentiation between K. kingae and MRSA osteoarthritis.
We should like to thank the reviewer very much for his interest and his thoughtful comments on our paper. His comments have stimulated a careful new reading of our paper, and this has allowed us to improve the quality and the fluidity of our text.
Major comments
Introduction section, first sentence. In addition to the permanent sequelae of osteomyelitis (“bone development and function”), the authors should mention the long-term disabilities resulting from septic arthritis, such as limping, decreased motion range or ankylosis, joint instability, etc.
Corrected as suggested
Materials and Methods section, page 2, last paragraph. Details of the nucleic acid amplification test should be provided (universal 16S rRNA gene target, K. kingae-specific (rtx operon, groEL, mdh genes) ?
A paragraph about microbiological methods has been added in Materials and Methods and provides complete details about nucleic acid amplification tests used in this study.
Materials and Methods section. Small bones and joints and intervertebral discs, which are not easily accessible and rarely subjected to a surgical aspiration or biopsy, were involved in a few patients. Were oropharyngeal specimens obtained for the molecular diagnosis of K. kingae, as advocated by the authors in previous publications [Pediatrics 2013; 131: e230-e235], [J. Child. Orthop. 2010; 4:173-175.], [Pediatr. Infect. Dis. J. 2013; 32:1296-1298]?
Yes, this point has been added in the section Materials and Methods.
Discussion section, page 12, first paragraph, lines 13-15. The authors suggest that the thrombocytosis observed in K. kingae osteoarthritis patients may result from a long-standing infection evolution induced by a low-grade virulence pathogen. Although K. kingae osteomyelitis frequently has an insidious onset and the disease is diagnosed with delay (after 9.2 ± 9.4 days after onset of symptoms), septic arthritis caused by the organism usually has an acute onset, and the infection is suspected after an average of only 3.2 ±3.0 days [Pediatr. Infect. Dis. J. 2010; 29:639-643].
The reviewer has right, but the hypothesis of which explain reactive thrombocytosis associated with K. kingae makes sense. OAIs caused by K. kingae generally follow a mild clinical course, probably behaving like a long-standing infection, which can lead to a more sustained reactive thrombocytosis.
Minor comments
Abstract line 3 and Introduction line 3. Replace "discovery: with "detection".
Corrected as suggested.
Introduction: line 13. Replace "pathogens responsible" with "causative pathogens" to avoid repetitions.
Corrected as suggested.
Introduction section, last sentence. The article compares the clinical presentation and laboratory markers of only two bacterial species (K. kingae and methicillin-susceptible S. aureus. The sentence should read: "The present study aimed to compare the signs and symptoms of disease caused by K. kingae and methicillin-susceptible Staphylococcus aureus (MSSA) and then establish predictive values to enable the differentiation of OAIs induced by these two organisms".
Corrected as suggested.
Figure 1. The actual number of patients in the different age-group categories should be added.
There are inconsistencies in the nomenclature of the two bacterial species. Kingella kingae is referred to as "K. Kingae" (no Italics, Figure 2 and 3) and "K. Kingella" (in Figure 2 Legend), and Staphylococcus aureus as "MSSA", "S. Aureus" (Figure 3, Legend, Figure 3, Legend).
Corrected as suggested. For more fluidity, Figures 2, 3 and 4 were merged in Figure 2 (A, B, C).
Reviewer 2 Report
This is a nice paper regarding the comparison of clinical and biological characteristics between S.aureus and K. Kingae osteoarticular infections in a large pediatric series. Although the paper has an advantage based on the number of patients and the time period covered, final results confirm the already known K. Kingae OAI picture. Despite the lack of novelty I find the paper comprehensible, scientifically solid and with a good review of the current literature in the discussion session.
I suggest the following points to be included/corrected:
1.Consider adding more data regarding the microbiological confirmation of either pathogen. Authors should include a paragraph in results ( or a table) regarding culture (including fluid inoculated in blood culture vials) and molecular methods results. In addition in materials and methods the molecular method used should be defined (multiplex PCR, single PCR targeting for S. aureus and K. Kingae) and referenced.
2. In materials and methods section, line 100: Number of WBC irrespective of age?Authors should provide reference for these infection parameters.
2. Results: Authors should avoid duplication in text and Table 1 regarding inflammatory markers' values. Preferably include only the table. I suggest the same amendment for the duplication between the results in the last paragraph and figure 5.
3. There is no data for the microbiological confirmation of either pathogen. Authors should include a paragraph in results ( or a table) regarding culture (including fluid inoculated in blood culture vials) and molecular methods results. In addition in materials and methods the molecular method used should be defined (multiplex PCR, single PCR targeting for S. aureus and K. Kingae) and referenced.
4. Figure 1 should be replaced so the age distribution will be more clear and less confusing (preferably with bars at each age)
5. In discussion (lines 303-305) it is mentioned that there is no significant difference regarding WBC counts between MSSA and K.Kingae infections. However in table 1 a significant difference is shown. Authors should explain this discrepancy.
6. Perhaps a second comparison in terms of inflammation markers between KK and MSSA OAI should be done for age-matched patients from both groups.
Author Response
This is a nice paper regarding the comparison of clinical and biological characteristics between S.aureus and K. Kingae osteoarticular infections in a large pediatric series. Although the paper has an advantage based on the number of patients and the time period covered, final results confirm the already known K. Kingae OAI picture. Despite the lack of novelty I find the paper comprehensible, scientifically solid and with a good review of the current literature in the discussion session.
Thank you very much for your review and for your constructive comments. We have tried to improve the text considering your propositions.
I suggest the following points to be included/corrected:
1.Consider adding more data regarding the microbiological confirmation of either pathogen. Authors should include a paragraph in results ( or a table) regarding culture (including fluid inoculated in blood culture vials) and molecular methods results. In addition in materials and methods the molecular method used should be defined (multiplex PCR, single PCR targeting for S. aureus and K. Kingae) and referenced.
A paragraph about microbiological methods has been added in Materials and Methods and provides complete details about nucleic acid amplification tests used in this study.
We added also more data regarding the microbiological confirmation of either pathogen in the results’ section.
- In materials and methods section, line 100: Number of WBC irrespective of age? Authors should provide reference for these infection parameters.
We treat this paradox of WBC counts in the discussion for explaining the results of this study (especially the fact that WBC inferior to 17,000 cells/mm3 are considered normal in children below 4 years old).
- Results: Authors should avoid duplication in text and Table 1 regarding inflammatory markers' values. Preferably include only the table. I suggest the same amendment for the duplication between the results in the last paragraph and figure 5.
Corrected as suggested.
- There is no data for the microbiological confirmation of either pathogen. Authors should include a paragraph in results ( or a table) regarding culture (including fluid inoculated in blood culture vials) and molecular methods results. In addition in materials and methods the molecular method used should be defined (multiplex PCR, single PCR targeting for S. aureus and K. Kingae) and referenced.
We added a paragraph in results with complete data for explaining how the confirmation of either pathogens was realized, with the results of molecular methods results.
In addition, a paragraph about microbiological methods has been added in Materials and Methods and provides complete details about nucleic acid amplification tests used in this study.
- Figure 1 should be replaced so the age distribution will be more clear and less confusing (preferably with bars at each age)
We sincerely prefer this type of graphical representation than those with bars at each age. We would like to keep this one….
- In discussion (lines 303-305) it is mentioned that there is no significant difference regarding WBC counts between MSSA and K.kingaeinfections. However in table 1 a significant difference is shown. Authors should explain this discrepancy.
We wanted to say that there was probably no significant difference in WBC counts between OAIs caused by K. kingae or MSSA, when the cutoffs of normality are established according to the patients’ age. The sentence has been clarified.
- Perhaps a second comparison in terms of inflammation markers between KK and MSSA OAI should be done for age-matched patients from both groups.
We tried to do it but this analysis could not be done since the two populations are totally different with regard to the age of the patients.
Reviewer 3 Report
Many thanks to the authors and the editor for the possibility to review the manuscript "Osteoarticular infections in children: accurately distinguishing between MSSA an Kingella kingae".I would like to make some comments to the authors in order to improve the manuscript:
I do not quite understand how the sample was selected. Perhaps there could be a selection bias. Some cases were selected since 1997 and others since 2007. Perhaps all cases should have been selected since 2007. Were all OAIs taken into account? Perhaps some before 2007 were for KK and were not diagnosed. The explanation of how the sample was obtained and why the method does not induce any bias should be improved.
The methodology does not indicate that a multivariate logistic regression (or other statistical methodology) is performed, yet the authors conclude that they have designed a predictive model. How is this possible? The only thing that is provided are the AUC of certain individual values of the elements that would make up the model, not the AUC values of the model.
The authors should implement in the methodology of the study how they have constructed the model and the AUC and other parameters of the model. Otherwise their conclusions are not true, they would only provide the value of each variable analysed independently, not a model.
It is advisable to implement a table summarising all patient characteristics. In the rows all the variables, qualitative and quantitative, and in the rows: total, KK group, SA group and "p". It would be a more complete table 1, being able to delete of t value and the CI. All acronyms and variable measurements (months, years, °C, ºC, ºK, etc. in the table footnote) must be explained.
Authors should indicate what they consider significant (p-value) in the methods, not in the discussion.
Authors are advised not to repeat the information in the table in the text.
AUC values should be reported with their confidence interval.
The discussion repeats multiple numerical data already presented in the results. The authors should remove these numerical data from the discussion.
The conclusion would have to be rewritten since, as indicated above, a productive model has not been realised. The literature is old, only 4 articles are from the last decade, the rest are from before 2012 (25). There are relevant articles on this topic that the authors have not taken into account. By way of example:
DOI: 10.3390/microorganisms10061233
DOI: 10.1016/j.artmed.2020.101895
The authors should review the bibliography.
Author Response
Many thanks to the authors and the editor for the possibility to review the manuscript "Osteoarticular infections in children: accurately distinguishing between MSSA and Kingella kingae".I would like to make some comments to the authors in order to improve the manuscript:
We should like to thank the reviewer very much for his interest and his relevant comments on our paper.
I do not quite understand how the sample was selected. Perhaps there could be a selection bias. Some cases were selected since 1997 and others since 2007. Perhaps all cases should have been selected since 2007. Were all OAIs taken into account? Perhaps some before 2007 were for KK and were not diagnosed. The explanation of how the sample was obtained and why the method does not induce any bias should be improved.
OAIs caused by S. aureus were recruited since 1997. This date corresponds to the moment at which electronic medical charts have started in our hospital.
For OAIs due to K. kingae, the recruitment started when single PCR targeting for K. kingae was available in our hospital (2007)
The methodology does not indicate that a multivariate logistic regression (or other statistical methodology) is performed, yet the authors conclude that they have designed a predictive model. How is this possible? The only thing that is provided are the AUC of certain individual values of the elements that would make up the model, not the AUC values of the model.
The authors should implement in the methodology of the study how they have constructed the model and the AUC and other parameters of the model. Otherwise their conclusions are not true, they would only provide the value of each variable analyzed independently, not a model.
The reviewer is fully right. The univariate and multivariate logistic regression was added in this revised version (Table 3). This model aims to predict K. kingae in patients aged less than 48 months based on clinical and biological features.
It is advisable to implement a table summarizing all patient characteristics. In the rows all the variables, qualitative and quantitative, and in the rows: total, KK group, SA group and "p". It would be a more complete table 1, being able to delete of t value and the CI. All acronyms and variable measurements (months, years, °C, ºC, ºK, etc. in the table footnote) must be explained.
Modified as suggested
Table 1 was update with an additional column for all patients characteristics. All acronyms were also explained in the table footnote.
Authors should indicate what they consider significant (p-value) in the methods, not in the discussion.
Corrected as suggested
Authors are advised not to repeat the information in the table in the text.
Correct as suggested.
AUC values should be reported with their confidence interval.
Corrected as suggested
The discussion repeats multiple numerical data already presented in the results. The authors should remove these numerical data from the discussion.
We do not agree with this specific point; the discussion focuses on quantified criteria and comparisons with the data present in the literature require that the numerical data be mentioned and compared. Subtracting them would make the discussion difficult to follow
The conclusion would have to be rewritten since, as indicated above, a productive model has not been realized. The literature is old, only 4 articles are from the last decade, the rest are from before 2012 (25). There are relevant articles on this topic that the authors have not taken into account. By way of example:
DOI: 10.3390/microorganisms10061233
This paper has been added to the discussion. It was published in 2022 after the redaction of this paper.
DOI: 10.1016/j.artmed.2020.101895
However, this one does not correspond in any way with the treated argument.
The authors should review the bibliography.
Done as suggested
Reviewer 4 Report
In this study, the authors compare the signs and symptoms of OAIs due to K. kingae to those due to S. aureus. In my opinion, the study is interesting, and the conclusion is supported by the results, but I have several concerns that should be addressed, as follows:
- Materials and methods
- It is unclear how S. aureus and K. kingae are being diagnosed in these cases. Please add some details about the methods used for organism identification.
- Statistical analysis: Please specify how the optimal cut-off values have been determined.
- Results:
- There are several numerical results in the text. Please have these results tabulated to avoid duplication of the numerical results between the tables and the text.
- Table 1: Please specify the unit of measurement for each parameter in the table.
- A ROC curve analysis was performed, but we still need to add the ROC curve graph for each parameter.
- Multivariate logistic regression analysis should be performed, with the dependent variable being the group studied and the covariates being the parameters investigated. The significant variables should be combined using binary logistic regression, and their combined diagnostic accuracy should be determined using the ROC curve.
- Abbreviations should be defined in full terms when they first appear in the text, such as MSSA, MRSA, etc.
Author Response
In this study, the authors compare the signs and symptoms of OAIs due to K. kingae to those due to S. aureus. In my opinion, the study is interesting, and the conclusion is supported by the results, but I have several concerns that should be addressed, as follows:
We should like to thank the reviewer very much for his interest and his thoughtful comments on our paper.
- Materials and methods
- It is unclear how S. aureus and K. kingae are being diagnosed in these cases. Please add some details about the methods used for organism identification.
A paragraph about microbiological methods has been added in Materials and Methods and provides complete details about nucleic acid amplification tests used in this study.
- Statistical analysis: Please specify how the optimal cut-off values have been determined.
The univariate and multivariate logistic regression was added in this revised version (Table 3). This model aims to predict K. kingae in patients aged less than 48 months based on clinical and biological features.
- Results:
- There are several numerical results in the text. Please have these results tabulated to avoid duplication of the numerical results between the tables and the text.
Corrected as suggested
- Table 1: Please specify the unit of measurement for each parameter in the table.
Corrected as suggested
- A ROC curve analysis was performed, but we still need to add the ROC curve graph for each parameter.
Added as suggested
- Multivariate logistic regression analysis should be performed, with the dependent variable being the group studied and the covariates being the parameters investigated. The significant variables should be combined using binary logistic regression, and their combined diagnostic accuracy should be determined using the ROC curve.
The univariate and multivariate logistic regression was added in this revised version (Table 3). This model aims to predict K. kingae in patients aged less than 48 months based on clinical and biological features.
We added in the description how the significant variables have been combined using binary logistic regression, and we explain and how diagnostic accuracy has been determined using the ROC curve.
- Abbreviations should be defined in full terms when they first appear in the text, such as MSSA, MRSA, etc.
Corrected as suggested
Reviewer 5 Report
The manuscript by Benoit Coulin et al. describes the study of clinical and biological parameters of children with osteoarticular infections. The objective is to prognose the bacterial etiology (MSSA or K. kingae).
Revisions are needed.
English needs to be revised. Prefer passive turns of phrase. Acronyms must be introduced before their first use. Italicize "i.e.", "vs." and names of bacteria. Numbers less than or equal to 12 should be spelled out.The layout of Microorgansims should be strictly followed.
I think that the difference in inclusion period between MSSA and KK is a bias that should be taken into account, prefer to include data only from superimposable periods to avoid classification bias.
Table 1: p value and 95%CI are redundant.
In view of all the results obtained on Age/Temperature/CRP elevation/WBC/count/platelet count/ ESR, the authors should reanalyze their data to create a predictive score cumulating them. In addition, an AUROC seems to be a relevant factor to be determined.
Author Response
The manuscript by Benoit Coulin et al. describes the study of clinical and biological parameters of children with osteoarticular infections. The objective is to prognose the bacterial etiology (MSSA or K. kingae).
Revisions are needed.
English needs to be revised. Prefer passive turns of phrase.
English has been revised by a professional editor…
Acronyms must be introduced before their first use. Italicize "i.e.", "vs." and names of bacteria.
Corrected as suggested
Numbers less than or equal to 12 should be spelled out.
??? I have never seen such a rule
The layout of Microorgansims should be strictly followed.
Corrected as suggested
I think that the difference in inclusion period between MSSA and KK is a bias that should be taken into account, prefer to include data only from superimposable periods to avoid classification bias.
We did not consider it as a bias since before the use of PCR assay it was impossible for us to cultivate K. kingae. We have explained clearly in the new version the reasons for which the periods were not superimposable.
Table 1: p value and 95%CI are redundant.
Asked by another reviewer ….
In view of all the results obtained on Age/Temperature/CRP elevation/WBC/count/platelet count/ ESR, the authors should reanalyze their data to create a predictive score cumulating them. In addition, an AUROC seems to be a relevant factor to be determined.
We do not agree with this proposition. In fact, The AUC assessed using ROC curves for temperature at admission, CRP and differential WBC count were all significantly different from the non-informative value of 0.5. The AUC for age at diagnosis demonstrated an excellent ability (AUC = 0.91) to discriminate between the two populations. The AUCs for CRP (0.79), temperature at admission (0.76) and platelet count (0.76) indicated a fair accuracy in discriminating between the two populations. The discriminatory accuracy between the two subgroups of patients was poor for WBC count (AUC = 0.62) and fail for ESR (AUC = 0.58). Therefore, the cut-offs defined at WBC ≥ 9,350/mm3 and ESR ≥ 51.5 mm/h could not be used as discrimination tools.
Round 2
Reviewer 3 Report
I thank the authors for their interest in answering and implementing the suggestions made to them in the paper.
Author Response
We thank the reviewer for considering that his suggestions have been taken into consideration and that our responses are in line with his expectations.
Reviewer 4 Report
The authors have adequately addressed all my queries.
Author Response
We are glad to note that our answers are inline with the expectations of the reviewer.
Reviewer 5 Report
Tha authors answered most, but not all my previous comments.
I think that the difference in inclusion period between MSSA and KK is a bias that should be taken into account, prefer to include data only from superimposable periods to avoid classification bias.
We did not consider it as a bias since before the use of PCR assay it was impossible for us to cultivate K. kingae. We have explained clearly in the new version the reasons for which the periods were not superimposable.
--> Even if the authors describe the reason, I cannot agree with them. It would be clearer to compare strictly the same period, as numerous factors (not bacterial but societal for example) could deeply impact the results. Please consider.
Table 1: p value and 95%CI are redundant.
Asked by another reviewer ….
--> Choose one or the other.
In view of all the results obtained on Age/Temperature/CRP elevation/WBC/count/platelet count/ ESR, the authors should reanalyze their data to create a predictive score cumulating them. In addition, an AUROC seems to be a relevant factor to be determined.
We do not agree with this proposition. In fact, The AUC assessed using ROC curves for temperature at admission, CRP and differential WBC count were all significantly different from the non-informative value of 0.5. The AUC for age at diagnosis demonstrated an excellent ability (AUC = 0.91) to discriminate between the two populations. The AUCs for CRP (0.79), temperature at admission (0.76) and platelet count (0.76) indicated a fair accuracy in discriminating between the two populations. The discriminatory accuracy between the two subgroups of patients was poor for WBC count (AUC = 0.62) and fail for ESR (AUC = 0.58). Therefore, the cut-offs defined at WBC ≥ 9,350/mm3 and ESR ≥ 51.5 mm/h could not be used as discrimination tools.
--> Excluding WBC and ESR, a score that include CRP, temperature at admission, age at diagnosis could improve discrimantory accuracy in order to build the best model possible with your data. Please consider.
Author Response
The authors answered most, but not all my previous comments.
I think that the difference in inclusion period between MSSA and KK is a bias that should be taken into account, prefer to include data only from superimposable periods to avoid classification bias.
We did not consider it as a bias since before the use of PCR assay it was impossible for us to cultivate K. kingae. We have explained clearly in the new version the reasons for which the periods were not superimposable.
--> Even if the authors describe the reason, I cannot agree with them. It would be clearer to compare strictly the same period, as numerous factors (not bacterial but societal for example) could deeply impact the results. Please consider.
ANSWER :
First, we think it is important to clarify this point with this reviewer. We did not deliberately choose different inclusion periods. It turns out that before the implementation of PCR assays in our hospital, we were unable to highlihght Kingella kingae. The reaserch was done for both groups during the same period but it was totally unefficient for Kingella before having pcr assays.
In addition, we do not understand how inclusion of subjects from slightly different time periods can lead to classification biais. We do not see how societal factors may deeply impact the results…. ! And we will ask him which exposure may change the outcome in this specific study?
The only element that could have created a bias of classification had been a difference of MRSA implication between the two periods, which was not the case !
The aim of this study is just to compare the signs and symptoms of disease caused by K. kingae and methicillin-susceptible Staphylococcus aureus (MSSA) and then establish predictive values to enable the differentiation of OAIs induced by these two organisms.
We did not plan to study incidence or prevalence of both these infections.
Finally, we would like to ask the reviewer which classification bias are affecting this study to his opinion ? Non-differential or differential biais ?
To our knowledge, classification bias occurs when variables that affect outcomes are inadequately recorded at the beginning of a study.
This is not the case in our study.
In retrospective observational studies, researchers must take into consideration groups of participants that had a risk factor or exposure and its association to some outcome. Detecting bias in classification should thus ask the following question: could the outcomes(s) be biased based on how participants were classified in the study due to their risk factor/exposure? What are risk factors or exposure that maycan legally conduct to classification biais ?
But none of this applies to our study !
Table 1: p value and 95%CI are redundant.
Asked by another reviewer ….
--> Choose one or the other.
ANSWER :
As suggested by the reviewer, the sentence describing the p value has been removed
In view of all the results obtained on Age/Temperature/CRP elevation/WBC/count/platelet count/ ESR, the authors should reanalyze their data to create a predictive score cumulating them. In addition, an AUROC seems to be a relevant factor to be determined.
We do not agree with this proposition. In fact, The AUC assessed using ROC curves for temperature at admission, CRP and differential WBC count were all significantly different from the non-informative value of 0.5. The AUC for age at diagnosis demonstrated an excellent ability (AUC = 0.91) to discriminate between the two populations. The AUCs for CRP (0.79), temperature at admission (0.76) and platelet count (0.76) indicated a fair accuracy in discriminating between the two populations. The discriminatory accuracy between the two subgroups of patients was poor for WBC count (AUC = 0.62) and fail for ESR (AUC = 0.58). Therefore, the cut-offs defined at WBC ≥ 9,350/mm3 and ESR ≥ 51.5 mm/h could not be used as discrimination tools.
--> Excluding WBC and ESR, a score that include CRP, temperature at admission, age at diagnosis could improve discrimantory accuracy in order to build the best model possible with your data. Please consider.
ANSWER:
We do not understand this remark, since this is exactly what was done.
We invite therefore the reviewer to read again the description of thr statistical method and the results.
In fact, we defined six predictors (i.e. age, temperature, WBC, CRP, ESR and platelet count) and used non-parametric receiver operating characteristic (ROC) curves to assess their abilities to differentiate between patients with K. kingae OAIs and those with S. aureus OAIs. The areas under the ROC curve (AUC) and their 95% confidence intervals were assessed using the nonparametric method.
Cut-off values were determined for all six parameters, and only those with a discriminative ability > 0.75 were selected for further analysis. The parameters were then dichotomised and determined for each patient. The proportions of OAIs due to K. kingae and S. aureus were assessed according to the number of parameters present at admission.
In order to predict regardless of age OAIs caused by K. kingae in children less than 4 years old, the clinical and laboratory parameters (temperature at admission, CRP, WBC, ESR, Platelet count) were then included in a univariate and multivariate logistic regression model, for which adjusted OR and 95% CI were calculated.
The AUC assessed using ROC curves for temperature at admission, CRP and differential WBC count were all significantly different from the non-informative value of 0.5. The AUC for age at diagnosis demonstrated an excellent ability (AUC = 0.91) to discriminate between the two populations. The AUCs for CRP (0.79), temperature at admission (0.76) and platelet count (0.76) indicated a fair accuracy in discriminating between the two populations. The discriminatory accuracy between the two subgroups of patients was poor for WBC count (AUC = 0.62) and fail for ESR (AUC = 0.58). Therefore, the cut-offs defined at WBC ≥ 9,350/mm3 and ESR ≥ 51.5 mm/h could not be used as discrimination tools.
Based on our results, the best predictive model for a K. kingae OAI included the following parameter cut-offs: age < 43 months, temperature at admission < 37.9°C, CRP < 32.5 mg/L and platelet count > 361,500/mm3.